# The Genetic and Immunologic Landscape Underlying the Risk of Malignant Progression in Laryngeal Dysplasia

**DOI:** 10.3390/cancers15041117

**Published:** 2023-02-09

**Authors:** Francesco Chu, Fausto Maffini, Daniela Lepanto, Davide Vacirca, Sergio Vincenzo Taormina, Rita De Berardinis, Sara Gandini, Silvano Vignati, Alberto Ranghiero, Alessandra Rappa, Susanna Chiocca, Massimo Barberis, Marta Tagliabue, Mohssen Ansarin

**Affiliations:** 1Division of Otolaryngology and Head and Neck Surgery, IEO, European Institute of Oncology IRCCS, Via Ripamonti 435, 20141 Milan, Italy; 2Division of Pathology, IEO, European Institute of Oncology IRCCS, Via Ripamonti 435, 20141 Milan, Italy; 3Department of Experimental Oncology, IEO, European Institute of Oncology IRCCS, Via Adamello 16, 20139 Milan, Italy; 4Department of Biomedical Sciences, University of Sassari, 07100 Sassari, Italy

**Keywords:** laryngeal dysplasia, tumor microenvironment, immunity system, laryngeal cancer, tumor-infiltrating lymphocytes, TILs, immune system

## Abstract

**Simple Summary:**

Our study investigates the alterations of the dysplastic micro-environment and immune system that contribute to the progression from laryngeal dysplasia to invasive cancer. As known, laryngeal dysplasia could represent the initial premalignant lesions for the development of laryngeal squamous cell cancer. Thus, the possibility to differentiate between progressing (PDy) and non-progressing dysplasia (NPDy) is essential for intensive surveillance programs and cancer prevention.

**Abstract:**

(1) Background: The development of laryngeal cancer is a multistep process involving structural alterations of the epithelial mucosa, from dysplasia (LDy) to invasive carcinoma. In this study, we define new biomarkers, prognostic for malignant transformation, in patients affected by LDy. (2) Methods: We used targeted next-generation sequencing and immunohistochemical analysis to define the mutational and immunological landscape of 15 laryngeal dysplasia progressing to invasive cancer (progressing dysplasia), as well as 31 cases of laryngeal dysplasia that did not progress to carcinoma (non-progressing dysplasia). Two pathologists independently analyzed the presence of tumor-infiltrating lymphocytes in LDy pre-embedded paraffin-fixed specimens. The RNA-based next-generation sequencing panel OIRRA was used to evaluate the expression of 395 genes related to immune system activation. (3) Results: High TILs are significantly correlated with a higher risk of malignant transformation. The non-brisk pattern was significantly associated with an 86% reduced risk of malignant progression (OR = 0.16, 95% CI: 0.03–0.5, *p* = 0.008). TILs showed a highly positive correlation with CCR6, CD83, HLA-DPB1, MX1 and SNAI1, and they were inversely correlated with CD48, CIITA, CXCR4, FCER1G, IL1B, LST1 and TLR8. (4) Conclusions: TILs have a great potential to identify high-risk progression dysplasia and thus to define surveillance protocols and prevention programs.

## 1. Introduction

Laryngeal squamous cell cancer (LSCC) represents one-third of all head and neck cancers and is burdened by significant morbidity and mortality for advanced-stage disease. About 200,000 new cases of laryngeal cancer were diagnosed worldwide in 2017, and recent studies show that the incidence and prevalence have increased by 12.0% and 23.8%, respectively, during the past three decades [1,2].

The development of LSCC is a multistep process involving structural alterations of the epithelial mucosa, from initial premalignant lesions (low/high-grade dysplasia) to invasive carcinoma [3,4].

From a clinical point of view, laryngeal dysplasia (LDy) is observed as leukoplakia of the vocal folds at the fiberoptic examination of the larynx [5]. Several authors have reported their experiences and management of small cohorts of patients, and even the recent data from the literature define patients with LDy as an inhomogeneous group for whom the grade of dysplasia alone is an insufficient prognostic factor for the development of laryngeal cancer. Thus, it is not possible to distinguish between lesions at high risk or low risk of malignant progression [6,7,8].

Although a general agreement has been lacking among multidisciplinary fields for the management of these lesions, the ability to differentiate between high-risk and low-risk LDy is crucial for an intensive surveillance program and early diagnosis followed by adequate treatment.

Recently, we have highlighted the role of some clinical prognostic factors other than the histopathological findings that have a crucial significance for LDy transformation: the supraglottic involvement, the multifocality and a history of recurring LDy [7].

Furthermore, in recent years, it has been reported that apart from the tumor genetic factors, the host immune response and the tumor-promoting functions of cellular components in the LSCC tumor microenvironment and extracellular matrix significantly contribute to cancer progression for advanced disease [9,10].

Based on this background, it seems legitimate to ask ourselves if an alteration of the dysplastic microenvironment and immunologic system plays a role in the progression from LDy to invasive cancer. To address this issue in a comprehensive manner, we used targeted next-generation sequencing and immunohistochemical analysis to elucidate the mutational and immunological landscape of 15 cases of laryngeal dysplasia progressing to invasive cancer (progressing dysplasia; PDy), as well as 31 cases of laryngeal dysplasia that did not progress to carcinoma (non-progressing dysplasia, NPDy).

## 2. Materials and Methods

We performed a retrospective review of all patients treated with transoral laser microsurgery (TLM) between January 2005 and December 2020 in the Department of Otolaryngology and Head and Neck Surgery of a tertiary comprehensive cancer centre. All patients were referred to transoral laser microsurgery with radical intent according to the current international standards of care [11,12]. At the final histopathological reports, we considered only patients with a first diagnosis of LDy. All patients were followed up with laryngoscopy every six months for the first two years and annually thereafter and were instructed to anticipate or require further visits in the case of voice alterations. In the case of malignant progression (LSCC), all patients were discussed with the multidisciplinary board and referred to treatments according to the current standards of care.

Data collected were as follows: age; sex; past medical history; pre-operative smoke and alcohol habits; site of the laryngeal lesion; surgical procedure adopted; histopathological findings and severity of the squamous intraepithelial neoplasia (SIN) [7], and progression to LSCC during follow up.

Patients were excluded from the current study in the case of the following:-Previous history of invasive laryngeal cancer-Previous laryngeal surgical procedures or treatments-Immunitary diseases or concurrent medical treatments interfering with the immunitary system function (i.e., sclerodermia, rheumatoid arthritis, steroid immunosuppressive treatments, etc.)

To address the immunitary system factors and immunogenetic alterations prognostic for malignant progression in a comprehensive vision, we evaluated the presence of stromal tumor-infiltrating lymphocytes (TILs) in the specimen samplings collected and stored after surgery. Moreover, we used the RNA-based next-generation sequencing (NGS) panel Oncomine Immune Response Research Assay (OIRRA) (ThermoFisher, Waltham, MA, USA) (13) to measure the expression of genes associated with lymphocyte regulation, cytokine signaling, lymphocyte markers, and checkpoint pathways in all the cases of PDy as well as two matched pair NPDy per each case.

All patients signed an informed consent form to use data for scientific purposes, and the work was carried out in accordance with the Declaration of Helsinki (ID Hospital trial:2519).

### 2.1. Histopathological Evaluation of TILs

Two different pathologists (F.M. and D.L.) independently analyzed the presence of TILs. After the specimen sampling, a formalin-fixed paraffin-embedded (FFPE) slide was prepared and stained with hematoxylin and eosin (H&E).

Fifteen patients had more than one episode of LDy, always treated with TLM according to the standards of care. In such cases, we performed the immunohistochemical analysis on the LDy specimen collected from the first surgical procedure. Thus, the final pool included 46 LDy specimens, of which 31 were NPDy and 15 PDy.

For TILs evaluation, we considered the connective tissue below the dysplasia and counted the lymphocytes, plasma cells, and macrophages in one mm2 area. Granulocytes, neutrophils, eosinophils, basophils, and connective and endothelial cells were excluded from the count.

The microscope used was a Leica DMRB with 20/10 ocular at 20X of objective for both pathologists (F.M./D.L.). The count was made with a cell counter for both pathologists. For each slide, they reported the highest, lowest, and the average count of TILs in the analyzed slide defined respectively as High-TILs, Low-TILs and Average count. TILs were evaluated in a 1 mm thin border at the edge of the dysplasia and stroma in one mm2 area.

The two skilled pathologists evaluated the type of distribution of TILs in the connective tissue. For “homogenous distribution”, we referred to equally distributed TILs along the epithelial–stromal border, while for “heterogenous distribution”, we referred to irregularly distributed TILs along the epithelial–stromal border due to the presence of some areas with more concentrated infiltrates and other with poorly represented or absence of TILs. The two skilled pathologists were asked to report even the Brisk/non-Brisk characterization of TILs. The term “B” refers to the amount of active and vigorous TILs along the basal lamina and specifically involving the basal lamina and the epithelial cells along the epithelial–stromal border as described for melanoma [11,13]. “NB” TILs represent the infiltrate distributed only focally and not along the entire epithelial–stromal border and without or minimal involvement of epithelial cell along the epithelial–stromal border, while “B” TILs define lymphocytes that infiltrate the border diffusely (Figure 1a–d) [14,15,16].

### 2.2. Genetic Analysis

Of the 46 LDy specimens, we obtained successful RNA transcript in 24 specimens, 9 PDy and 15 NPDy. In such cases, the RNA-based next-generation sequencing panel OIRRA (TermoFisher, Waltham, MA, USA) was used to evaluate of the expression of 395 genes related to immune system activation, such as genes associated with lymphocyte regulation, cytokine signaling, lymphocyte markers, checkpoint pathways and tumor characterization. The RNA extraction was automatically performed with the Promega Maxwell instrument (Promega, Madison, WI, USA) using the Promega Maxwell RSC RNA FFPE kit and quantified with the Quantus fluorometer (Promega, Madison, WI, USA). Real-time PCR was used for RNA quality assessment. RNA library preparation and chip loading were automatically realized on the Ion Chef System (TermoFisher, Waltham, MA, USA), and the sequencing step was run using the Ion S5 System (TermoFisher, Waltham, MA, USA). The targeted RNA sequencing analysis is obtained with the Torrent Suite Immune Response RNA plugin that produces gene transcript data automatically.

### 2.3. Statistical Analysis

Demographic and clinical characteristics were presented with descriptive statistics by PDy status, which is the main outcome measure.

#### 2.3.1. TILs Analysis

We evaluated the agreement between the two pathologists by Bland–Altman plots, investigating the correspondent 95% Limits of Agreement (LoA) of the mean difference.

Differences in TIL measurements between patients with PDy and NPDy were evaluated with univariate Wilcoxon Rank Tests. We also investigated the association of TILs with PDy by logistic regression models, considering the average values of the two pathologists’ evaluations.

#### 2.3.2. Gene Expression Analysis

RNA-sequencing data were obtained for the gene expression level evaluation. Gene expression Read Per Million (RPM) data were centered log-ratio transformed after a non-parametric multivariate imputation of zeros for compositional data [17]. After the exclusion of one gene with low variability, data from 399 genes were available for this analysis. First, gene expression levels were categorized considering the presence or absence of expression. Second, gene expression levels were classified as “high” or “low” expression considering a level of expression higher and lower/equal to the sample median gene expression, respectively. Lastly, gene expression was considered on a continuous scale.

A heatmap was generated by performing a sparse Partial Least Square-Differential Analysis (sPLS-DA) (sevenfold cross-validation and 100 repeats) and selecting the most discriminative genes by using the first and second component loading vectors by using MixOmics Package 6.18.1. In particular, sPLS-DA was evaluated by applying sevenfold cross-validation with 100 repeats.

We furtherly investigated the correlations between gene expressions and TILs by Spearman rank correlation coefficients.

## 3. Results

The final cohort analyzed includes 46 patients. Thirty-nine patients were current or former smokers, while no patient had a current or previous history of alcohol abuse/addiction. No statistically significant differences were found for patients’ tumor characteristics (Table 1).

Similar to our previous results [7], both smoke and alcohol habits did not result statistically associated with the risk of progression to invasive LSCC.

### 3.1. TILs Analysis

When evaluating TIL density by PDy/NPDy status, PDy specimens showed to be characterized by a significantly higher amount of inflammatory infiltrates along the basal lamina compared to NPDy. These differences were appreciated by both examiners (Table 2).

When considering the level of interaction between the immunitary system and the dysplasia due to intrinsic characteristics of the inflammatory infiltrate, the Brisk pattern, as evaluated by the first pathologist, was found to be significantly associated with an 86% reduced risk of malignant progression (OR = 0.16, 95% CI: 0.03–0.5, *p* = 0.008). This result was independently confirmed by the second pathologist, who found that the Brisk pattern was associated with an 89% reduced risk of malignant progression (OR = 0.11, 95% CI: 0.02–0.43, *p* = 0.003).

Finally, concerning the double-blind system used to evaluate TILs, the two examiners showed a good agreement index, when evaluating both “Brisk” and “Density” (“Brisk” Cohen Kappa = 0.64, “Density” Cohen Kappa = 0.60) (Figure 2).

### 3.2. Gene Expression Analysis

As shown in the heatmap (Figure 3), the Sparse Penalized Last Square Discriminant Analysis (S-PLSDA) characterized the expression of 28 genes, which were able to discriminate between PDy and NPDy patients (FCER1G, SRGN, CCR4, HLA-DQA1, CA4, CD63, ITGAM, IL2RB, CD83, TNFRSF17, TIGIT, MS4A1, ITGB1, HLA-F-AS1, LILRB2, IDO2, JCHAIN, NECTIN2, MPO, AKT1, CD79A, IKZF1, LST1, TAP1, LRG1, CD27, STAT6, CD33).

Finally, at the Spearman Rank Test, TIL density was found to be highly positively correlated with gene expression of CCR6, CD83, HLA-DPB1, MX1, SNAI1 and inversely correlated with CD48, CIITA, CXCR4, FCER1G, IL1B, LST1 and TLR8 (Rho > 0.5).

## 4. Discussion

From our preliminary results, TILs showed to be prognostic for malignant progression even at a precancerous stage, not only as a median value of density but either as low and high values, thus permitting to differentiate NPDy from PDy.

Interestingly, the immune response was significantly more elicited by PDy than NPDy. Such evidence was also reported by others investigating mucosal dysplasia of the oral cavity and respiratory system [18] and reporting a more represented immune infiltration into the epithelial compartment for high-risk dysplasia, compared to low-risk dysplasia, thus suggesting that the pattern of lymphocyte infiltration is associated with the dysplastic severity. Similar to other experiences, it might be allegedly assumed that even for LDy, high-risk lesions exert a more intense anti-tumorigenic effect (increasing the density of TILs), and conversely, NPDy shares a different immunogenetic response characterized by the Brisk pattern of infiltration and low-density TILs.

In the present pilot study, the grade of dyplasia seems to be related to the NPDy/PDy status, but such results did not show to be clinically supported by our experience in a larger retrospective cohort of patients [7], yet confirming that still today the “grade” of LDy has a debated predictive value in Literature [6,7,8]. For NPDy and PDy, TILs change quantitatively from low density to high density and qualitatively from Brisk to non-Brisk, similar to what was reported for melanoma. More specifically, during the vertical growth phase of melanoma, Brisk TILs correlate with a better prognosis than non-Brisk TILs, thus stratifying patients with a low and high risk of metastasis [13,19,20].

From our preliminary results, we could assume that, independently of the histopathological grading, the risk of progression from dysplasia to invasive cancer could be predicted on the base of the Quantitative TILs (QTILs) response. This is true even if we consider the quality of the inflammatory cells. More specifically, increasing QTILs is related to shifted Qualitative TILs (QuTILs) from Brisk to non-Brisk response for NPDy and PDy, respectively.

The non-Brisk quality of TILs for PDy might hint at a weak interaction between the immune system and the dysplastic cells. Similarly, this feature was reported as an important poor prognostic value in melanoma progression with metastasizing properties [13,19,20].

In advanced LSCC, high TILs were associated with a better prognosis, independently of the Brisk/non-Brisk pattern. Conversely, at a precancerous stage, TILs seem to increase quantitatively and decrease qualitatively in PDy, as also reported by others [18].

We might assume that during cancerogenesis, PDy causes high QTILs and shows progressing oncogenetic alterations leading to non-Brisk-QuTILs, similar to what was reported for melanoma [13,19,20].

High-density TILs were found to be positively correlated with CCR6, CD83, HLA-DPB1, MX1, SNAI1 and inversely correlated with CD48, CIITA, CXCR4, FCER1G, IL1B, LST1 and TLR8.

C-C motif-chemokine Receptor-6 (CCR6) belonged to a chemokine receptor family and was found to be expressed in memory T-cells and even immature dendritic cells, macrophage receptor in B-lineage cells and correlates with recruitment and maturation of dendritic, macrophage and T-cells during many immunological responses. CCR6 can increase the interleukina-17 (IL-17) level, helping to recruit helper T-cells and the regulatory T-cells [21,22].

CD83 encodes for a type I transmembrane protein belonging to an immunoglobulin receptor family. It seems to be involved in antigen presentation, binding to immature and mature dendritic cells, and it inhibits presentation of the antigen to T lymphocytes [23].

This receptor works together with other major histocompatibility complex II (MHC-II) proteins. The MHC, Class II, DP Beta 1 (HLA-DPB1) is a protein-coding gene complex, expressed by dendritic cells on the plasma membrane surface. This receptor presents the antigen on the cell surface through endocytosis and degradation of the antigen peptides after phagocytosis, using the antigens presenting cell route. When the antigens are presented, a subset of activated T-lymphocytes (CD4 + T-Cells) binds and recognizes the antigen. From our preliminary results, it seems that the expression of CD83 and HLA-DPB1 plays a pivotal role in the T-lymphocytes’ immunitary response.

An upregulation of CD83 in the dendritic cells might modulate the MHC-II presentation, thus increasing the recruitment of both immature and mature dendritic cells amplifying the immunitary response against the tumor [23].

In our results, the upregulation of HLA-DPB1 and CD83 seems to improve the recruitment of activated CD4 and T lymphocytes and increase the ability to recognize the neoplastic cells.

This mechanism should favor a more aggressive response against neoplastic cells but, in long-lasting disease, lead to an exhausted and poor-quality response of CD4 + T-lymphocytes and promotes malignant progression over time.

MX Dynamic Like GTPase 1, or Myxovirus resistance 1 (MX1), is a protein-coding gene that resulted upregulated in other solid tumors as infiltrating breast cancer, melanoma, prostatic adenocarcinoma, and even an independent poor prognostic factor in head and neck squamous cell carcinoma [24,25]. The upregulation of MX1 is induced by interferon-alpha due to the presence of tumoral and dendritic cells.

In our study, high TILs might be associated with PDy due to an augmented IFN-1 secretion and upregulation of MX1 for its anti-apoptotic activity.

Snail Family Transcriptional Repressor 1 (SNAI1) is involved in the epithelial–mesenchymal transition (EMT), providing cells with staminal cell features [26] and the ability to differentiate in many other cellular lineages [27] with mesenchymal and epithelial phenotypes. EMT transition was observed to be associated with an upregulation of SNAI1 overall in an early phase of cancerogenesis [28]. Olmeda et al. reported an upregulation of SNAI1 in tumor growth and progression. Conversely, a growth reduction and differentiation were observed when the tumor was silenced for SNAI1 [29].

C-X-C Motif Chemokine Receptor 4 (CXCR4) belongs to a chemokine receptor family.

Knockdown of CXCR4 decreases the migration and invasion of neoplastic cells due to a reduction in the activity of Matrix metallopeptidase 9 (MMP-9) and 13 (MMP-13). MMP-9 and MMP-13 are important for the degradation of the basal layer membrane and the extracellular matrix, thus favoring tumor infiltration and metastases [30].

SNAI1 and CXCR4 might be targeted for medical therapy in patients affected by dysplasia [31,32].

CD48 is an immunoglobuline-like receptor belonging to a signaling-lymphocyte-activation molecule (SLAM) superfamily [33]. CD48 supports the lytic activity of cytotoxic T-lymphocyte against cancer growth [34].

The Class II MHC Transactivator (CIITA) is an MHC II trans-activating factor supporting CD4 + T-Lymphocyte recruitment; thus, the downregulation of CIITA was reported to be related to aggressive and metastatic carcinoma [35]. This feature was also reported in LSCC [36].

Fc Epsilon Receptor Ig (FCER1G) was found to be downregulated in our cases. FCER1G binds with many FcR alpha-chain receptors, with IgE-FcR on the cellular surface. It recognizes the neoplastic antigen and drives an innate-like immune response [37]. The decrease in FCER1G in our cases can be due to a “deterioration” of the innate-like immune response [37].

Interleukin 1 Beta (IL-1B) is a cytokine involved in the inflammatory response and several cellular activities, such as proliferation, differentiation, and apoptosis. IL-1B promotes T-cell activation, antibodies secretion by B-cells, fibroblastic activity, and collagen production [38,39,40,41]. IL-1B was reported to be upregulated in head and neck SCC with drug resistance to erlotinib [42]. In our work, IL-1B was found to be downregulated in laryngeal precancerosis. Thus, our results could provide the basis for future studies on LDy medical treatments. The down-regulation of IL-1B and therapies with anti-EGFR chemotherapies should be considered for dysplastic lesions, as also reported by Yin et al. [42,43].

Leukocyte Specific Transcript (LST1) is a protein that inhibits lymphocyte proliferation; its role in cancerogenesis and drug resistance is poorly understood [44]. In our work, we hypothesize that low LST1 level drives neoplastic modifications on epithelial and mesenchymal cells, thus modifying the epithelial–mesenchymal transition.

Toll-Like receptor 8 (TLR8) belongs to the Toll-like receptor family, driving antigen recognition, innate immunitary response, and antibody-dependent mediated toxicity [45]. TLR8 is associated with oral SCC carcinogenesis. It acts concurrently with IL-1B and enhances the antineoplastic activity of NK cells in patients treated with cetuximab [46]. The stimulation of DCs due to TLR8 alone increases the CD80+ and CD83+ lymphocyte activation. This effect is even amplified when TLR8 is associated with cetuximab [46]. Low levels of TLR8 in our cases were shown to be related to an augmented risk of malignant progression in PDy.

Overall, from our analysis, different genes are involved in the immunitary response, including genes coding for membrane receptors, histocompatibility, immune modulators and immune checkpoints inhibitors. The different expression of these genes varies according to the different and very complex immunitary system response, thus influencing the quantitative and qualitative TILs.

Being a pilot study on a small cohort of patients, the genetic results from OIRRA might only hint at a correlation between some genes more or under-expressed in LDy and TILs, yet providing the basis for future studies investigating the mechanisms by which such alterations directly influence TILs and cancerogenesis, hence the malignant transformation from LDy to LSCC.

Finally, the evaluation of TILs, as described in our study, might be an easy and reproducible immunohistochemical test for risk stratification in patients affected by LDy. This aspect is fundamental for tailored surveillance programs based on TIL Density and Brisk/non-Brisk Pattern, thus permitting to reach a higher rate of LSCC early diagnosis and less invasive treatments.

Our pilot study certainly shows some important limitations due to its retrospective nature and the small cohort of patients. Considering the small number of patients and the high number of variables to be tested, our preliminary results were not significant at the Bonferroni test. They will be validated in further larger prospective cohort studies.

## 5. Conclusions

Nowadays, LSCC is still burdened by a poor prognosis for advanced-stage disease, and, independently of the oncological outcomes, even curative treatments often result in a worsening of a patient’s quality of life due to vocal and swallowing impairments. With these premises, there is a clear need for new biomarkers that might improve the rate of early diagnosis of early-stage LSCC, thus permitting minimally invasive and function-sparing laryngeal treatments with consequent better oncological and functional results.

From our pilot study, TILs emerged as an independent prognostic factor that permits a better risk stratification of patients affected by LDy. More specifically, high TILs and non-Brisk patterns showed to be correlated with a higher risk of LSCC.

This significant result throws important perspectives on future tailored surveillance programs, proposing that the inflammatory response plays a pivotal role, even at a precancerous stage and shows great potential in the early detection of patients at higher risk of malignant progression.

## Figures and Tables

**Figure 1 cancers-15-01117-f001:**
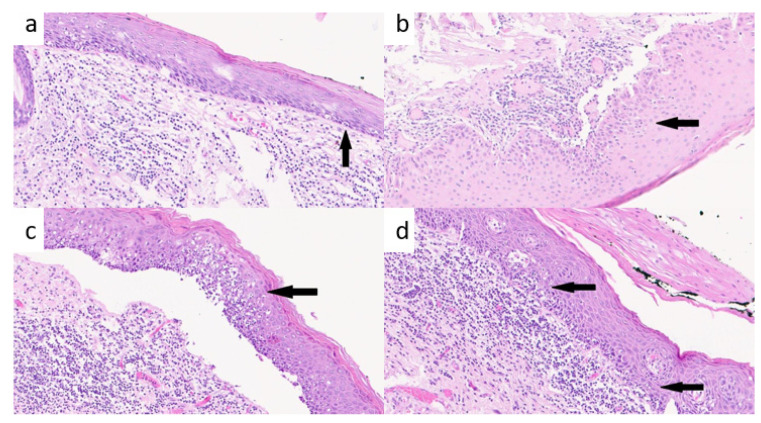
(**a**) Non-Brisk Low Grade: low-grade dysplasia with a reduced lymphocyte infiltrate in the connective tissue and few lymphocytes along the epithelial border (black arrow); (**b**) Non-Brisk Low Grade: low-grade dysplasia with detached epithelium from connective tissue and few lymphocytes along the epithelial border (black arrow); (**c**) Brisk Low Grade: low grade dysplasia with detached epithelium from the connective tissue and a rich lymphocytes infiltrate along the epithelial border (black arrow); (**d**) Brisk Low Grade: low grade dysplasia with squamous epithelium along the border of the connective tissue (black arrows) and a rich lymphocytes infiltrate in the stroma.

**Figure 2 cancers-15-01117-f002:**
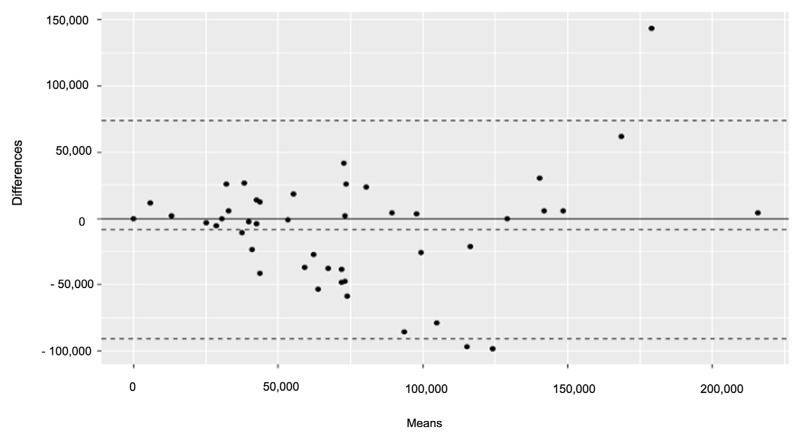
Bland–Altman plot of the two inter-observers in the counting system for TILs. The inter-observer differences in the counting system tend to increase for TILs higher than 75,000.

**Figure 3 cancers-15-01117-f003:**
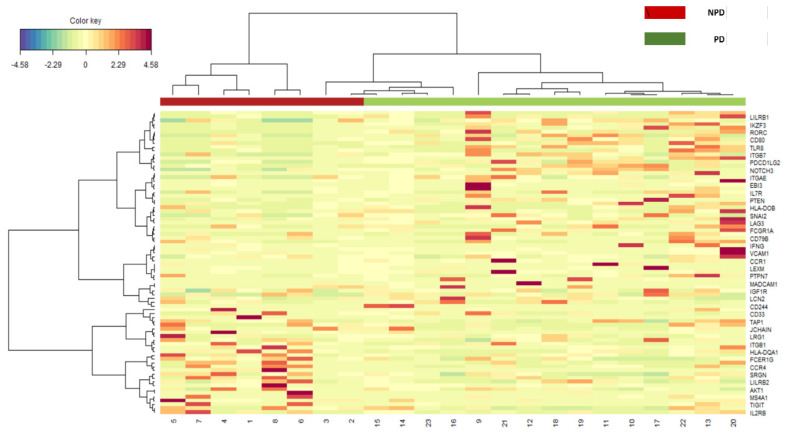
S-PLS discriminant analysis genes between progressing (PDy) and non-progressing (NPDy) dysplasia.

**Table 1 cancers-15-01117-t001:** Patients’ characteristics by non-progressing dysplasia/progressing dysplasia status.

Characteristics	NPDy n = 31 (100%)	PDy n = 15 (100%)	*p*-Value ^1^
**Sex**			0.2
Female	8 (26%)	1 (6.7%)	
Male	23 (74%)	14 (93%)	
**Age, Median (IQR)**	58 (54, 64)	66 (56, 70)	0.2
**Smoking**			0.8
Never	4 (13%)	3 (20%)	
Current	13 (42%)	6 (40%)	
Former	14 (45%)	6 (40%)	
**Cancer site**			0.6
Glottis	27 (87%)	13 (87%)	
Supraglottis	2 (6.5%)	0 (0%)	
Glottis + Supraglottis	2 (6.5%)	2 (13%)	
**Histological Grade**			0.029
mild SIN	11 (35%)	3 (20%)	
Intermediate SIN	10 (32%)	1 (6.7%)	
severe SIN	10 (32%)	11 (73%)	
**Multifocal**			0.4
No	26 (84%)	11 (73%)	
Yes	5 (16%)	4 (27%)	

^1^*p*-value: Fisher’s exact or Chi-square test or Wilcoxon rank sum test for the continuous variable. IQR: interquartile range. NPDy: non-progressing dysplasia; PDy: progressing dysplasia. SIN: squamous intraepithelial neoplasia.

**Table 2 cancers-15-01117-t002:** Pathological features by non-progressing dysplasia/progressing dysplasia status.

Characteristics	NPDy n = 31 (100%)	PDy n = 15 (100%)	*p*-Value ^2^
**Examiner 1**			
**Distribution ^1^**			0.3
Homogeneus	1 (3.2%)	2 (13%)	
Eterogeneus	28 (90%)	13 (87%)	
Not evaluable (NE)	2 (6.5%)	0 (0%)	
**Brisk ^1^**			0.011
No	4 (13%)	8 (53%)	
Yes	25 (81%)	7 (47%)	
Heterogenous-NE	2 (6.5%)	0 (0%)	
**High-TILs ^1^**	69,268 (46,175–103,503)	105,892 (72,054–190,685)	0.022
**Low-TILs ^1^**	18,312 (11,146–39,411)	47,771 (30,652–101,114)	0.001
**Average Count ^1^**	49,363 (29,855–69,665)	74,841 (47,969–147,890)	0.009
**Density 20** **X ^1^**	34,634 (23,089–51,752)	52,946 (36,027–95,342)	0.019
**Density 20X2 ^1^**	9156 (4976–19,705)	23,885 (15,326–50,557)	0.001
**Average Density ^1^**	24,283 (14,928–34,833)	37,420 (23,985–73,945)	0.008
**Examiner 2**			
**Distribution ^1^**			0.5
Homogeneus	28 (90%)	15 (100%)	
Not evaluable (NE)	3 (9.7%)	0 (0%)	
**Brisk ^1^**			0.002
No	5 (16%)	10 (67%)	
Yes	23 (74%)	5 (33%)	
Heterogenous-NE	3 (9.7%)	0 (0%)	
**High-TILs ^1^**	97,930 (54,936–151,672)	160,032(103,105–203,025)	0.028
**Low-TILs ^1^**	15,127 (9554–33,439)	71,656 (27,468–86,783)	0.005
**Average Count ^1^**	60,549 (33,503–90,764)	107,484 (69,068–142,118)	0.011
**Density 20X ^1^**	48,965 (27,468–75,836)	80,016 (51,552–101,513)	0.031
**Density 20X/2 ^1^**	7564 (4777–15,724)	35,828 (13,734–43,392)	0.004
**Average Density ^1^**	26,871 (17,217–45,382)	53,742 (34,534–71,059)	0.011

^1^ n (%); Median (IQR); ^2^
*p*-value: Fisher’s exact or Chi-square test or Wilcoxon rank sum test for the continuous variable. NPDy: non-progressing dysplasia. PDy: progressing dysplasia.

## Data Availability

The data presented in this study are available on request from the corresponding author. The data are not publicly available due to ethical reasons.

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
