# Peer review of "The Genetic and Immunologic Landscape Underlying the Risk of Malignant Progression in Laryngeal Dysplasia"

_cancers, 2023, doi:10.3390/cancers15041117_

Round 1
Reviewer 1 Report
First of all, I would like to express great praise to the authors for their valuable scientific contribution.
The abstract should contain the number of patients or the sample size. Please add this to the abstract.
The authors give a good introduction to the topic of laryngeal dysplasias. I am still missing the clear definition of progressing and non-progressing dysplasia here. please add this definition and also the literature reference.
Furthermore, please name the immunological diseases that are a criterion for exclusion or give examples.
How many had recurrent dysplasia? Or was that also an exclusion criterion?
Also here the definitions of NPDy and PDy is missing, the latter is the main outcome measure. Please add them. What is your surveillance protocoll and follow-up?
Please add in line 92 how many specimen samplings were collected and examined.
I do not find the results complete. In Table 1 includes SIN, which does not appear in the text. Complete the description of SIN in this section. Does alcohol have an effect on PDy status?
Further, readers may find it extremely confusing that the first section of the results or TILs analysis deals exclusively with the interrater agreement and the TILs characterization falls short. Please add this in the main text!
In table 2 many abbreviations are missing, please add them in the legend.
Overall, there are some errors in the text (e.g. trasformation line 66) that are easy to avoid, please correct.
In the discussion you should describe the transition from NPDy to PDy. Also discuss your surveillance protocol. Where do you see the quantitative TILs cutoff value? In line 228-229 authors write about high and low responder, is this your observation? How was this measured? Is there any literature on this observation?
Overall, the discussion is not fluid. There are sections about tumor suppressor genes strung together, this is where the text should have coherence and a grand finale. Please add the limitations here!
Concerning the conclusion, lines 329 and 330 with references belong in the discussion. The conclusions are inadequate. The authors have to re-write this section.
Author Response
First of all, I would like to express great praise to the authors for their valuable scientific contribution.
1 - The abstract should contain the number of patients or the sample size. Please add this to the abstract.
We corrected the abstract according to the reviewer’s requests
“ Methods: We used targeted next-generation sequencing and immunohistochemical analysis to define the mutational and immunological landscape of 15 cases of laryngeal dysplasia progressing to invasive cancer (progressing dysplasia), as well as 31 cases of laryngeal dysplasia that did not progress to carcinoma (non-progressing dysplasia)..”
2 - The authors give a good introduction to the topic of laryngeal dysplasias. I am still missing the clear definition of progressing and non-progressing dysplasia here. please add this definition and also the literature reference.
We modified the introduction, specifying the PDy/NPDy definitions
“…To address this issue in a comprehensive manner, we used targeted next-generation sequencing and immunohistochemical analysis to elucidate the muta-tional and immunological landscape of 15 cases of laryngeal dysplasia progressing to invasive cancer (progressing dysplasia; PDy), as well as 31 cases of laryngeal dysplasia that did not progress to carcinoma (non-progressing dysplasia, NPDy)…”
3 - Furthermore, please name the immunological diseases that are a criterion for exclusion or give examples.
We specified the immunological diseases that are a criterion for exclusion and gave examples.
“..immune diseases or concurrent medical treatments interfering with immune system function (i.e. scleroderma, rheumatoid arthritis, steroid immunosuppressive treatments, etc.)…”
4 - How many had recurrent dysplasia? Or was that also an exclusion criterion?
Histopathological evaluation of TILS
“…Fifteen patients had more than one episode of LDy, always treated with TLM accord-ing to the standards of care. In such cases, we performed the immunohistochemical analysis on the LDy specimen collected from the first surgical procedure. Thus the final pool included 46 LDy specimens, of which 31 were NPDy and 15 were PDy. ..”
5 - Also here the definitions of NPDy and PDy is missing, the latter is the main outcome measure. Please add them.
We addressed this issue in the Introduction Section (see point 2)
6 - What is your surveillance protocoll and follow-up?
In the “Materials and Methods” section we specified:
“…All patients were followed up with laryngoscopy every six months for the first two years and annually thereafter, and were instructed to bring forward or require further appointments if they experienced any voice alterations. In the case of malignant progression (LSCC), all patient cases were discussed by the multidisciplinary board and referred to treatments according to the current standards of care…”
7 - Please add in line 92 how many specimen samplings were collected and examined.
Histopathological evaluation of TILS
“…Fifteen patients had more than one episode of LDy, always treated with TLM according to the standards of care. In such cases, we performed the immunohistochemical analysis on the LDy specimen collected from the first surgical procedure. Thus, the final pool included 46 LDy specimens, of which 31 were NPDy and 15 were PDy...”
8 - I do not find the results complete. In Table 1 includes SIN, which does not appear in the text. Complete the description of SIN in this section.
In the “Materials and Methods” section we specified:
“…severity of the squamous intraepithelial neoplasia (SIN)… and progression to LSCC during follow up….”
And modified the Table 1 according to the previous used grading system
9 - Does alcohol have an effect on PDy status?
In the “Results” section we specified:
“…Similarly, to our previous results [7], both smoke and alcohol habits did not result sta-tistically associated with the risk of progression to invasive LSCC…”
10 - Further, readers may find it extremely confusing that the first section of the results or TILs analysis deals exclusively with the interrater agreement and the TILs characterization falls short. Please add this in the main text!
We modified the TILs results section:
“…when evaluating TILs density by PDy/NPDy status, PDy specimens were shown to be characterized by a significantly higher amount of inflammatory infiltrates along the basal lamina compared to NPDy. These differences were observed by both examiners (Ta-ble 2).
When considering the level of interaction between the immune system and the dys-plasia due to intrinsic characteristics of the inflammatory infiltrate, the brisk pattern as evaluated by the first pathologist was found to be significantly associated with an 86% reduced risk of malignant progression (OR=0.16, 95%CI: 0.03-0.5, P=0.008). This result was independently confirmed by the second pathologist who found that the brisk pattern was associated with an 89% reduced risk of malignant progression (OR=0.11, 95%CI: 0.02-0.43, P=0.003).
Finally, concerning the double-blind system used to evaluate TILs, the two examiners showed a good agreement index, when evaluating both "brisk" and "density" (“brisk” Co-hen Kappa = 0.64, “Density” Cohen Kappa = 0.60). (Fig. 2)…”
11- In table 2 many abbreviations are missing, please add them in the legend.
We better specified: 1 n (%); Median (IQR); 2 p-value: Fisher's exact or Chi-square test or Wilcoxon rank sum test for the continuous varia-ble. NPDy: non-progressing dysplasia. PDy: progressing dysplasia. NE: not evaluable. TILs: tumor-infiltrating lym-phocytes High-TILs: highest count of TILs per slide. Low-TILs: lowest count of TILs per slide. Average Count: average count of TILs per slide. Brisk: active and vigorous TILs along the basal lamina (No: non Brisk Pattern. Yes: Brisk Pattern)
12 - Overall, there are some errors in the text (e.g. trasformation line 66) that are easy to avoid, please correct.
We asked our Institutional native English Speaker, Mr William William Russell-Edu to revise the manuscript and updated the acknowledgment Section
“…Aknowledgement: The authors thank William Russell-Edu for revision of the English revision…”
14 - In the discussion you should describe the transition from NPDy to PDy. Also discuss your surveillance protocol.
We specified the definition of NPDy and PDy in the Introduction, Materials and Methods and Discussion.
Introduction:… To address this issue in a comprehensive manner, we used targeted next-generation se-quencing and immunohistochemical analysis to elucidate the mutational and immuno-logical landscape of 15 cases of laryngeal dysplasia progressing to invasive cancer (progressing dysplasia; PDy), as well as 31 cases of laryngeal dysplasia that did not progress to carcinoma (non-progressing dysplasia, NPDy)…”
Materials and Methods:… Finally we retrospectively selected 31 patients treated for LDy that didn’t fall as LSCC during follow up (NPDy) and 15 LDy patients that were treated afterwards for LSCC (PDy)
Discussion:… from our preliminary results, TILs showed to be prognostic for malignant progression even at a precancerous stage, not only as a median value of density but either as low or high values, thus permitting investigators to differentiate laryngeal dysplasia at higher risk of progression to invasive cancer (progressing dysplasia; PDy), from low risk dyspla-sia, not progressing to carcinoma (non-progressing dysplasia, NPDy)…
We also specified our surveillance protocol in the “materials and methods” section.
“…All patients were followed up with laryngoscopy every six months for the first two years and annually thereafter, and were instructed to bring forward or require further appoint-ments if they experienced any voice alterations. In the case of malignant progression (LSCC), all patient cases were discussed by the multidisciplinary board and referred to treatments according to the current standards of care…”
15 - Where do you see the quantitative TILs cutoff value?
The method for cell count and definition of High and Low-TILs was specified..
The count was made with a cell counter for both pathologists. For each slide, they reported the highest, lowest, and the average count of TILs in the analyzed slide, defined respectivley as High-TILs, Low-TILs and Avarage count. TILs were evaluated in a 1 mm thin border, at the edge of the dysplasia and stroma in a 1 mm2 area.
16 - In line 228-229 authors write about high and low responder, is this your observation? How was this measured? Is there any literature on this observation?
We thank the reviewer for his comment. Thus we better specified this TILs aspect related to the density and quality of immune response evaluated as “brisk/non-Brisk pattern”. Thus we modified the discussion accordingly
“…For NPDy and PDy, TILs change quantitatively from low density to high density and qualitatively from brisk to non-brisk, similar to that reported for melanoma. More specifi-cally, during the vertical growth phase of melanoma, brisk TILs correlate with a better prognosis than non-brisk TILs, thus stratifying patients with a low and high risk of me-tastasis [13, 19, 20].
From our preliminary results, we could assume that, independently of the histo-pathological grading, the risk of progression from dysplasia to invasive cancer could be predicted on the base of the Quantitative TILs (QTILs) response. This is true even if we consider the quality of the inflammatory cells. More specifically, increasing QTILs is re-lated to shifted Qualitative TILs (QuTILs) from brisk to non-brisk response for NPDy and PDy, respectively.
The non-brisk quality of TILs for PDy might hint at a weak interaction between the immune system and the dysplastic cells. Similarly, this feature was reported as an im-portant poor prognostic value in melanoma progression with metastasizing properties [13, 19, 20].
In advanced LSCC, high TILs were associated with a better prognosis, independently of the brisk/non-brisk pattern. Conversely, at a precancerous stage, TILs seem to increase quantitatively and decrease qualitatively in PDy, as also reported by other invesigators [18].
We might assume that during cancerogenesis, PDy causes high QTILs and shows progressing oncogenetic alterations leading to non-brisk-QuTILs, similar to that reported for melanoma [13, 19, 20]…”
17 - Overall, the discussion is not fluid. There are sections about tumor suppressor genes strung together, this is where the text should have coherence and a grand finale. Please add the limitations here!
Done
18 - Concerning the conclusion, lines 329 and 330 with references belong in the discussion. The conclusions are inadequate. The authors have to re-write this section.
We thank the reviewer for his comment (17 and 18) , thus we modified the discussion and conclusions.
“…Overall, from our analysis, different genes are involved in the immune response, in-cluding genes coding for membrane receptors, histocompatibility, immune modulators and immune checkpoint inhibitors. The different expression of these genes varies accord-ing to the different and very complex immune system response, thus influencing the quantitative and qualitative TILs.
Being a pilot study on a small cohort of patients, the genetic results from OIRRA might only hint at a correlation between some genes which are over- or underexpressed in LDy and TILs, thereby providing the basis for future studies investigating the mechanisms by which such alterations directly influence TILs and cancerogenesis, and thus the ma-lignant transformation from LDy to LSCC.
Finally, the evaluation of TILs, as described in our study, might be a straightforward and reproducible immunohistochemical test for risk stratification in patients affected by LDy. This aspect is fundamental for tailored surveillance programs based on TIL Density and brisk/non-brisk Pattern, thus permitting a higher rate of LSCC early diagnosis, and less invasive treatments.
Our pilot study shows certain important limitations due to its retrospective nature and the small cohort of patients. Considering the small number of patients and the high number of variables to be tested, our preliminary results were not significant at the Bon-ferroni test. They will be validated in further larger prospective cohort studies.
- Conclusion
LSCC is still burdened by a poor prognosis for advanced-stage disease, and, inde-pendently of the oncological outcomes, even curative treatments often result in a worsen-ing of patient's quality of life due to vocal and swallowing impairments. With these premises, there is a clear need for new biomarkers that might improve the rate of early di-agnosis of early-stage LSCC, thus permitting minimally invasive and function-sparing laryngeal treatments with consequent better oncological and functional results.
From our pilot study, TILs emerged as an independent prognostic factor that permits a better risk stratification of patients affected by LDy. More specifically, high TILs and non-brisk patterns were shown to be correlated with a higher risk of LSCC.
This significant result highlights important perspectives regarding future tailored surveillance programs, and leads us to propose that the inflammatory response exerts a pivotal role even at a precancerous stage, and shows great potential in the early detection of patients at higher risk of malignant progression….”
Reviewer 2 Report
I found the topic very interesting, and the analysis of the PDy and NPDy was certainly extensively done. I note several areas that were done extremely well, including the analysis of the degree of agreement and disagreement of the two expert pathologists. The genetic testing was quite comprehensive. The description of the patterns of TILs (Brisk and non-Brisk) , heterogeneity or lack thereof, and absolute number count were inventively done, and despite being a non-expert in the field, I felt the images and descriptions allowed me to understand these definitions.
I would state my criticisms of the paper in two sections.
1). My main issue with the paper relates to the explanation of the recruitment of so-called Progressive Dysplasia and non-Progressive Dysplasia. It seems that the 15 PDy cases were taken from TLM cases. It is never overtly explained, but it is alluded to that all these cases contained invasive cancer. Is that correct? In which case, does that describe the lesions as "progressive" compared to their previous status as only dysplasia? And is the previous presence of dysplasia (but not cancer) inferred from the fact that they are cancer now? Or, were there previous biopsies showing dysplasia but not cancer? Conversely, it is not really made explicit where the 21 cases of NPDy were generated from. Were these biopsies done at some point in the past, and was there clinical followup to ensure that the lesions did not progress? These are critical questions which require explicit answers, because the entire conclusions of the paper rely entirely on the division of cases into these two groups, but the methods of acquiring these cases and dividing cases into groups is never made explicit. It does not allow me to really judge the accuracy of the conclusions.
2). The conclusions do draw from the (strong) data generated by the paper, but as I state in issue 1, this is entirely dependent on the accurate division of cases into PDy and NPDy. However, I wonder with all these variables being evaluated if a Bonferroni correction should be considered.
Author Response
Reviewer 2
I found the topic very interesting, and the analysis of the PDy and NPDy was certainly extensively done. I note several areas that were done extremely well, including the analysis of the degree of agreement and disagreement of the two expert pathologists. The genetic testing was quite comprehensive. The description of the patterns of TILs (Brisk and non-Brisk) , heterogeneity or lack thereof, and absolute number count were inventively done, and despite being a non-expert in the field, I felt the images and descriptions allowed me to understand these definitions.
I would state my criticisms of the paper in two sections.
1). My main issue with the paper relates to the explanation of the recruitment of so-called Progressive Dysplasia and non-Progressive Dysplasia. It seems that the 15 PDy cases were taken from TLM cases. It is never overtly explained, but it is alluded to that all these cases contained invasive cancer. Is that correct? In which case, does that describe the lesions as "progressive" compared to their previous status as only dysplasia? And is the previous presence of dysplasia (but not cancer) inferred from the fact that they are cancer now? Or, were there previous biopsies showing dysplasia but not cancer? Conversely, it is not really made explicit where the 21 cases of NPDy were generated from. Were these biopsies done at some point in the past, and was there clinical follow up to ensure that the lesions did not progress? These are critical questions which require explicit answers, because the entire conclusions of the paper rely entirely on the division of cases into these two groups, but the methods of acquiring these cases and dividing cases into groups is never made explicit. It does not allow me to really judge the accuracy of the conclusions.
We thank the reviewer for his comments. Thus we better specified the definition of progressing and non progressing dysplasia in the introduction and discussion section.
Furthermore we specified in the materials and methods section that all cases were retrospectively selected on the base of the final histopathological reports of laryngeal dysplasia (LDy), from transoral laser microsurgery…and followed up according to the institutional protocol. Thus we retrospectively selected 31 patients treated for LDy that didn’t fall as LSCC during follow up (NPDy) and 15 LDy patients that were treated afterwards for LSCC (PDy)
Furthermore we modified the discussion and conclusions
2). The conclusions do draw from the (strong) data generated by the paper, but as I state in issue 1, this is entirely dependent on the accurate division of cases into PDy and NPDy.
See above
However, I wonder with all these variables being evaluated if a Bonferroni correction should be considered.
We better specified in the discussion: “…Our pilot study shows certain important limitations due to its retrospective nature and the small cohort of patients. Considering the small number of patients and the high number of variables to be tested, our preliminary results were not significant at the Bon-ferroni test. They will be validated in further larger prospective cohort studies…”
Round 2
Reviewer 2 Report
Thank you for addressing my concerns. There is a slight grammatical error at line 200.
Author Response
Thank you for the revision, we have corrected the slight grammatical error at line 200.